# Advances in Hydraulic Fracture Propagation Research in Shale Reservoirs

Xun Gong [1], Xinhua Ma [2,*], Yuyang Liu [2] and Guanfang Li [3]

1   School of Earth and Space Sciences, Peking University, Beijing 100871, China
2   Research Institute of Petroleum Exploration and Development, Beijing 100083, China
3   Institute of Geology and Geophysics, Chinese Academy of Sciences, Beijing 100029, China
*   Correspondence: xinhuam@petrochina.com.cn

**Abstract:** The characterization of artificial fracture propagation law in the fracturing process of shale reservoirs is the basis for evaluating the fracture conductivity and a key indicator of the reservoir stimulated effect. In order to improve the fracture stimulated volume of shale reservoirs, this paper systematically discusses the current status of research on artificial fracture propagation law from the research methods and main control factors and provides an outlook on its future development direction. The analysis finds that the study of fracture propagation law by using indoor physical simulation experiments has the advantages of simple operation and intuitive image, and the introduction of auxiliary technologies such as acoustic emission monitoring and CT scanning into indoor physical model experiments can correct the experimental results so as to better reveal the propagation mechanism of artificial fractures. At present, the numerical simulation methods commonly used to study the propagation law of artificial fractures include the finite element method, extended finite element method, discrete element method, boundary element method and phase field method, etc. The models established based on these numerical simulation methods have their own advantages and applicability, so the numerical algorithms can be integrated and the numerical methods selected to model and solve the different characteristics of the propagation law of artificial fractures in different regions at different times can greatly improve the accuracy of the model solution and better characterize the propagation law of artificial fractures. The propagation law of artificial fracture in the fracturing process is mainly influenced by geological factors and engineering factors, so when conducting research, geological factors should be taken as the basis, and through detailed study of geological factors, the selection of the fracturing process can be guided and engineering influencing factors can be optimized.

**Keywords:** fracture stimulated; artificial fracture; physical model experiment; numerical simulation

## 1. Introduction

The efficient development of shale gas is of great significance in alleviating energy shortages and optimizing energy structures and environmental protection. As the shale reservoir is characterized by ultra-low porosity and permeability, it is necessary to conduct fracturing stimulation on the shale reservoir to improve the porosity and permeability conditions of the reservoir so as to realize large-scale beneficial development of shale gas [1–3]. Therefore, the quality of reservoir reconstruction determines the development effect of shale gas in the later period. In order to maximize shale gas production, it is necessary to evaluate the effect of reservoir reconstruction. At present, 3D modeling, logging, seismic attribute analysis, reservoir description and other methods are commonly used to characterize the target reservoir, while the conductivity of complex fracture network is often used to evaluate the effect of reservoir reconstruction [4–7]. The more complex the network of artificial fractures formed during fracturing (this paper mainly explores the fracture propagation law of hydraulic fracturing, so the artificial fractures in this paper

are hydraulic fractures) and the stronger the conductivity, the better the effect of reservoir reconstruction [2,8,9] and the more conducive to the scale effect exploitation of shale gas. Up to now, many scholars have used indoor physical simulation test [10–13] and numerical simulation [14–17] and other methods to study the propagation law of artificial fractures and its influencing factors during fracturing and have used acoustic emission monitoring, CT scanning and other auxiliary technologies [18–21] to modify the research results so as to reveal the propagation law of hydraulic fractures.

Indoor physical simulation experiments simulate the study of hydraulic fracture extension process in shales deep underground by conducting hydraulic fracture tests on small-sized specimens, which enables the miniaturization of hydraulic fracture operations, and therefore many scholars have conducted numerous studies on the extension mechanism of hydraulic fractures using this method. Yuanfang Cheng et al. [9] Hou et al. [22], Guo et al. [23], and Shicheng Zhang et al. [24] investigated the effects of rock mechanical parameters, fracturing fluid, horizontal stress difference, and construction parameters on hydraulic fracture propagation using indoor fracturing physical simulation experiments simulations. Cheng et al. [25], Hengshuai et al. [26], Hou Zhengkun et al. [27], Fatahi et al. [28] have explored the influence of discontinuous structures, such as natural fractures and laminated surfaces, on the hydraulic fracture propagation pattern based on indoor physical model experiments and pointed out the criterion for discriminating hydraulic fractures penetrating natural fractures. In addition, some scholars, such as Zhenkun Hou [27], Shuai Heng [26], and Ma et al. [29], established a set of hydraulic fracture physical simulation and hydraulic fracture network characterization methods by summarizing previous research results and combining indoor physical simulation experiments with auxiliary techniques, such as acoustic emission monitoring and CT scanning, which greatly improved the reliability of indoor physical simulation experiments. Although the physical model experiments can characterize the real hydraulic fracturing operation, the obtained experimental results cannot truly characterize the subsurface hydraulic fracturing process because they are simplified for the complex subsurface conditions. Based on this, many scholars have established mathematical models of hydraulic fracture propagation [30–33] to investigate the propagation pattern of hydraulic fractures during fracturing by theoretical analysis. Two-dimensional fracture models were first established by Khristianovic and Zheltov [34] and Perkins and Kelerk [35] to analyze the propagation pattern of hydraulic fractures; subsequently, Geertsma and De Klerk [36], together with Daneshy [37,38] et al. and Nordgren et al. [39] modified the model and finally established the classical two-dimensional KGD and PKN fracture models. Simonson et al. [40] improved the two-dimensional model by introducing a stress intensity factor and established a proposed three-dimensional fracture model. Subsequently, Clifton and Abou-Sayed [41] established a three-dimensional model for hydraulic fracturing in infinitely large homogeneous isotropic elastic formations based on previous studies, which realized a three-dimensional hydraulic fracture model. The hydraulic fracture model achieved a breakthrough from two-dimensional to the proposed three-dimensional and a planar three-dimensional to a true three-dimensional, while A.V. Aksakov et al. [42] and A.V. Pestrikov [43] and other scholars [44] related proposed three-dimensional and planar three-dimensional models were validated, and software development for hydraulic fracture simulation was carried out, which accelerated the commercialization of fracture propagation models. With the rapid development of computer technology, more and more numerical simulation methods have been invoked to study the hydraulic fracture extension mechanism and its influencing factors. Advani [45], Rho et al. [46] established a finite element model based on the finite element method to characterize the vertical extension of fractures. Gordeliy et al. [47], Sheng et al. [48], Zou et al. [49] established a finite element model based on the extended Thallak et al. [50], Nagel et al. [51] used the discrete element method to establish a discrete fracture model with flow field coupling to simulate the propagation of hydraulic fractures in segmental fracturing. Olson [52], Bunger [53], Xu [54], Castonguay et al. [55] established a correlation model based on the boundary element method to analyze the characteristics of interseam

interference, fracture deflection, multi-fracture propagation and fracture geometry during fracture extension. According to these scholars, the influencing factors of hydraulic fracture extension can be divided into two categories: geological factors and engineering factors. Among them, geological factors [2,56–60] are determined by the nature of the shale itself, mainly the mineral composition of the shale, rock mechanical parameters, natural fractures, laminae surfaces, interseam interference, and ground stress. Engineering factors [11,61–64], on the other hand, are due to the influence of the artificially selected fracturing process during the extraction of shale gas, mainly the fracturing fluid, construction displacement, injection pressure and injection rate, and other factors. Both geological and engineering factors have a certain influence on fracture propagation, so when studying the mechanism of fracture extension, each influencing factor should be considered comprehensively to determine the degree of influence of each factor, clarify the main controlling factors of fracture propagation and achieve accurate prediction of fracture extension.

This paper firstly discusses the research methods of hydraulic fracture propagation in shale reservoir fracturing stimulation, based on the principles and research results of indoor physical simulation methods and numerical simulation methods; provides a comprehensive analysis of the applicability, advantages, and disadvantages of various methods; and analyzes the principles, applicability and research progress of the more commonly used mathematical models established based on numerical simulation methods. Then, the hydraulic fracture propagation, the influencing factors of hydraulic fracture propagation are discussed from both geological and engineering aspects, and the main controlling factors affecting fracture propagation are clarified. Finally, we provide a comprehensive analysis of the research methods and their influencing factors for hydraulic fracture propagation during shale fracturing and discuss how to improve various methods to achieve fine characterization of hydraulic fracture propagation.

## 2. Methodology for Studying Hydraulic Fracture Extension in Shale Reservoirs

As an important supplement to conventional oil and gas resources, shale gas has now become the main force of the world's oil and gas resource production growth. However, due to the dense structure of shale and its low pore and low permeability characteristics [1,3], it is determined that fracturing of shale reservoirs is required for shale gas development to improve the reservoir pore and permeability structure and thus obtain high production of shale gas [65]. Hydraulic fracturing technology [34,66–68] has now become the main technique for reservoir stimulation during shale gas development due to its advantages of low cost and low pollution. In order to obtain the largest possible volume of reservoir stimulation, it is essential to identify the extension pattern of hydraulic fractures during shale fracturing. Currently, the research methods [2,17,20,62,69–71] for hydraulic fracture extension mechanism are mainly indoor physical simulation experiments and numerical simulation methods and modified by using acoustic emission monitoring, CT scanning technology and post-compression fracture morphology observation so as to investigate the hydraulic fracture extension mechanism.

### 2.1. Indoor Physical Simulation Experiments

Indoor physical simulation experiments [11,61,69,71,72] are commonly used to study hydraulic fracturing operations in shale reservoirs by simplifying the subsurface situation and selecting materials with similar properties to shale or directly using the shale of the target layer to perform hydraulic fracturing experiments indoors using a true triaxial device (Figure 1) and combining acoustic emission monitoring, high-energy CT scanning, and post-compaction fracture morphology observation auxiliary techniques (Figure 2) was used to study the propagation process of hydraulic fractures during hydraulic fracturing in detail to understand the propagation mechanism of hydraulic fractures.

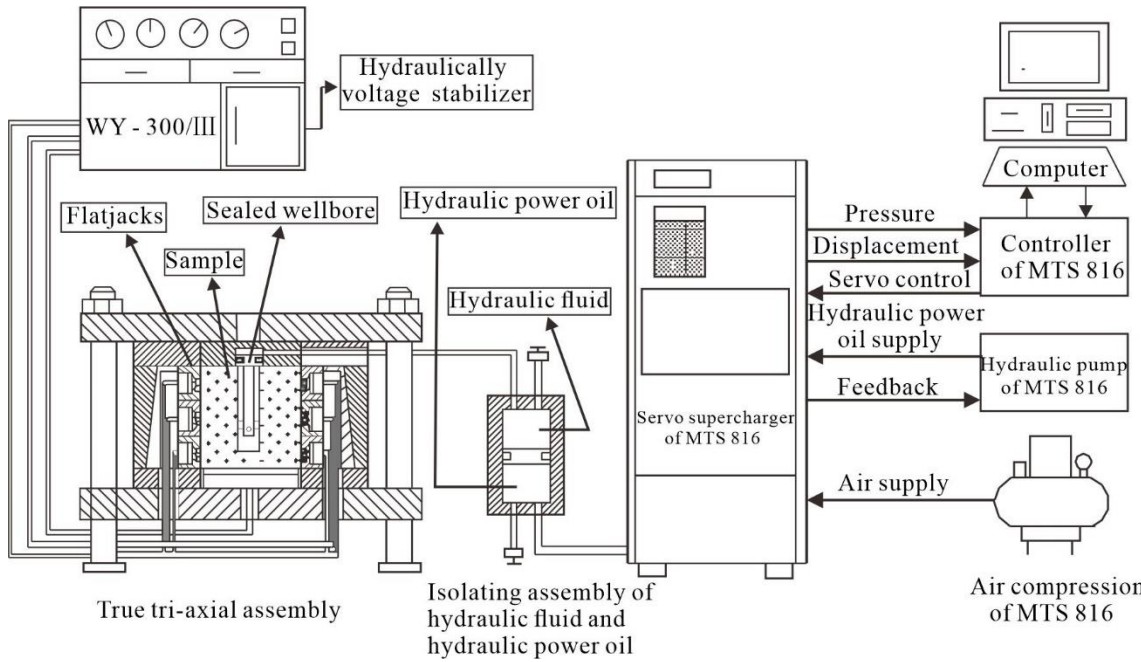

**Figure 1.** Laboratory physical simulation hydraulic fracturing experimental device system [58].

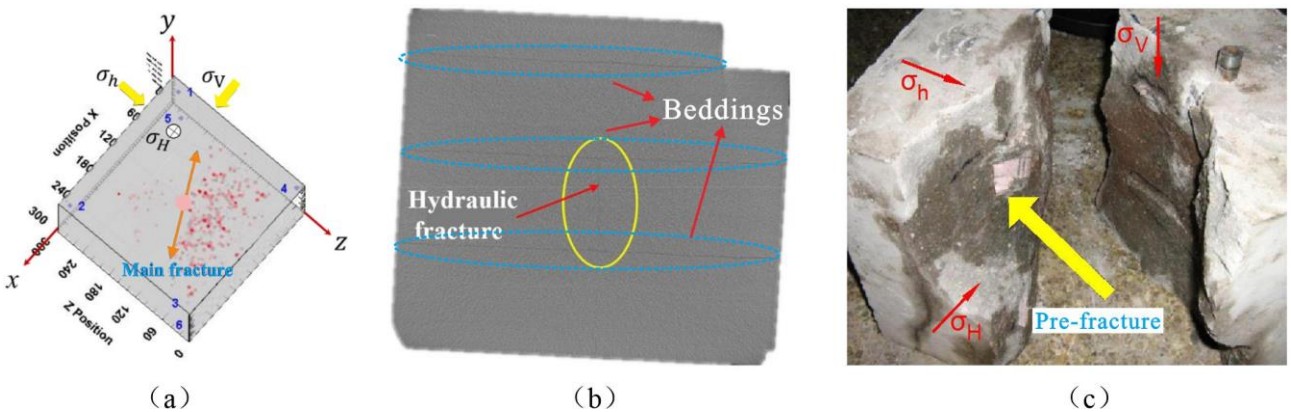

**Figure 2.** Fracture propagation study assistive technology [58,73,74]: (**a**) Acoustic emission monitoring technique, (**b**) CT scanning technique, (**c**) Post-pressure fracture morphology observation.

For the extension mechanism of hydraulic fractures during hydraulic fracturing, many researchers have investigated the propagation pattern of hydraulic fractures and their geometry during fracturing using indoor physical mode experiments. As early as 1963, Lamont et al. [75] investigated the effect of natural fractures on hydraulic fracture extension by preforming fractures in outcrop rocks and performing fracturing experiments on them, and they pointed out that the opening and direction of natural fractures do not affect the extension of hydraulic fractures. Blanton et al. [76] investigated the effect of natural fractures on hydraulic fracture extension by varying the angle of approach (defined as the angle between the natural fracture orientation and the maximum horizontal principal stress) and the horizontal stress difference (defined as the difference between the maximum horizontal principal stress and the minimum horizontal principal stress) was also analyzed for the interaction between the hydraulic fracture and the natural fracture. It was found that hydraulic fractures may pass through, bypass, turn, and cutoff when they encounter natural fractures. Warpinski [77] and Olson et al. [52,78] investigated the mutual interference between hydraulic fractures and natural fractures using indoor fracturing experiments and obtained similar results, and they pointed out that the geometry

of the extended fractures during fracturing is related to the horizontal stress difference, the fracturing fluid viscosity, the stress state and other factors. In addition to the influence of natural fractures on hydraulic fractures, many scholars have also considered the influence of cementation strength, laminar surface and the stress shadow effect on hydraulic fracture extension. Zhang Shicheng et al. [24] and Hou Bing et al. [22,79] found that the opening of weak surfaces, such as laminar surfaces, was beneficial to hydraulic fracture extension through indoor physical model experiments, and they pointed out that the deflection of the main fracture was related to the direction between the joint and the maximum principal stress. Hou et al. [8,22] combined hydraulic fracturing experiments with acoustic emission monitoring and CT scanning techniques to investigate the effects of parameters, such as laminar surfaces and fracturing fluid viscosity on hydraulic fracture extension. It was found that low viscosity fracturing fluid and low injection rate are beneficial to form complex fracture networks and increase reservoir stimulation volume under a high level of stress difference. Zhao et al. [11] analyzed the relationship between fracture extension and pumping parameters under constant pressure conditions based on hydraulic fracturing experiments and found that constant pressure injection could prolong the fracture extension period, and a higher constant pressure injection is beneficial to generate complex and curved hydraulic fractures. Many scholars have studied the hydraulic fracture propagation law using physical simulation experiments, however, due to the limited size of the samples used in physical simulation experiments, there is a "size effect" between the results of this test and the results of actual hydraulic fracturing operations. The "size effect" was investigated by de Pater et al. [80] and Bunger et al. [81] and the proportional relationship between the physical model test and the actual fracturing-related parameters was obtained, confirming that the "size effect" has a significant impact on the characterization results when using the physical model test results to characterize the actual fracturing operation. Therefore, it is important for researchers to identify the effect of "size" on the results of physical model tests. When the effect of "size effect" on the test results is clarified, a qualitative leap will be made in the physical model test. In addition, numerical simulations, acoustic emission monitoring, and CT scans are often used to correct the results of physical model tests so as to better characterize the extension mechanism of hydraulic fractures during fracturing and provide guidance for fracture optimization.

### *2.2. Numerical Simulation Methods*

Entering the 21st century, with the rapid development of computer technology and numerical calculation methods, numerical simulation methods have become an important tool in engineering practice and become an important method in the study of shale reservoir fracturing engineering. Currently, the numerical simulation methods used for the study of hydraulic fracture propagation mechanism in the shale fracturing process are finite element method, extended finite element method, discrete element method, boundary element method, phase field method, and other methods [2,13,14,32,63,69–71,82–84]. Among them, the first four are commonly used methods, while the phase field method is an emerging method that needs to be further studied, and based on these numerical methods, some models were established, such as the finite element model, extended finite element model, discrete fracture model, displacement discontinuity model, unconventional fracture model, etc. The mathematical models established were solved by using numerical simulation methods to realize the fine characterization of the hydraulic fracture propagation law during the fracturing process and to reveal the formation mechanism of the complex fracture network during the fracturing process.

### 2.2.1. Finite Element Method

The finite element method was originally used to solve continuity problems in complex engineering, such as the aerospace industry, and with the development of computer technology, the method was introduced into oil and gas development to study the propagation pattern of hydraulic fractures during fracturing. The basic idea of the finite element

method for fracture propagation is [2,13,14,17,69,71]: firstly, the discontinuous fracture is discretized with the fracture face as the boundary and the tip as the node, and the fracture is divided into several grid cells with a non-unique grid cell shape, and the local strain is used to identify the failure cells during fracture propagation so that the fracture caused by the interaction of hydraulic fracture and natural fracture, the propagation criterion, is hidden to realize the dynamic propagation of cracks (Figures 3a and 4b).

The traditional finite element method has more advantages in simulating fracture propagation during hydraulic fracturing of non-homogeneous shale under complex stresses, so many scholars have used it for the study of hydraulic fracture propagation law. Bao et al. [85] proposed a fully coupled finite element method for hydraulic fracturing by analyzing the hydraulic fracture propagation law under different energy dissipation mechanisms and filtration loss and introduced the cohesive method into it to reduce the size of the coupled equations. Zou et al. [49] introduced the cohesive unit into the finite element model based on the finite element method for studying the effect of natural fractures on hydraulic fracture propagation during fracturing. It was found that the complexity of the hydraulic fracture network decreases with the enhancement of the cementation strength of the natural fractures. Taleghani et al. [86] also added cohesive units to the finite element model to study the effect of natural fractures on the hydraulic fracture propagation and pointed out that the distribution and orientation of the natural fractures have an important influence on the geometry of the hydraulic fractures. Based on the previous studies and considering the influence of one or some factors on the hydraulic fracture propagation during the fracturing process, the above scholars improved the model accuracy by changing the set of control equations and introducing new methods and other measures to better characterize the hydraulic fracture propagation during the fracturing process using the finite element method.

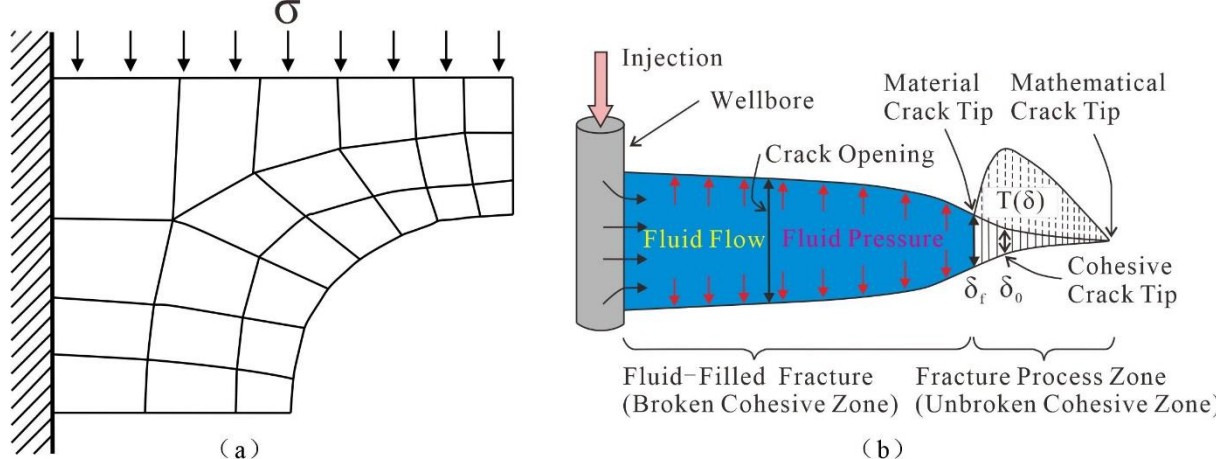

**Figure 3.** Simulation of hydraulic crack propagation by finite element method [28,87]: (**a**) Mesh generation by Finite Element Method (FEM); (**b**) Embedded cohesive zone in a hydraulic fracture, T, $\delta$, $\delta 0$ and $\delta_f$ represent traction tensor, displacement jump, and equivalently the separation and traction, respectively.

Many mathematical models based on the finite element method have been developed to study hydraulic fracture propagation, and one of the classic models is the cohesive unit model, which has been implemented in the commercial finite element software ABAQUS [49,85–90]. The traction separation criterion is chosen as the reservoir rock damage criterion in the fracture propagation process, i.e., the cohesive unit model characterizes the reservoir separately from the fracture by first dissecting the reservoir on a grid and then the cohesive unit model uses a set of independently divided grids to characterize the hydraulic fracture propagation process so that the propagation path of the hydraulic fracture is set in advance (Figure 3b). The cohesive unit model can be simulated without

generating a new mesh during the propagation process, which enhances its convergence and significantly improves the computational efficiency and is therefore widely used in the study of simple fracture propagation [13,14,32]. However, the propagation direction and path of the complex fracture network in the actual fracturing process are unknown, and the cohesive unit makes the fracture propagate only along the prefabricated path, the propagation path of the fracture is restricted, and the deflection problem of hydraulic fracture cannot be simulated [13,14], so it is also difficult for the cohesive model to simulate the formation process of the complex fracture net-work. Therefore, clarifying the propagation law of hydraulic fractures and their interaction mechanism with natural fractures has become the focus of hydraulic fracturing research. The introduction of the finite element method has deepened researchers' understanding of the fracture propagation mechanism, and to a certain extent, promoted the progress of the shale fracturing process. However, given the many challenges in fracture propagation, it is necessary to strengthen the research to improve the finite element method according to the characteristics of fractures, so that the method can better characterize the fracture propagation law.

### 2.2.2. Extend Finite Element Method

Since the traditional finite element method is difficult to reflect the dynamic propagation process of cracks when solving discontinuous problems, such as crack propagation, Belytschko et al. [91] proposed the extended finite element method. The extended finite element method [13,14,71,91] introduced a crack penetration unit and crack tip unit to characterize the discontinuous displacement field based on the traditional finite element method, and used the expanded shape function with discontinuous nature and the tip approximation displacement function to describe the existence and growth of cracks, respectively, making the crack and mesh interfaces independent of each other, which solved the drawback of the traditional finite element method in simulating fracture propagation that required mesh (Figure 4), which substantially improves the computational efficiency and enables effective simulation of dynamic processes, such as crack extension and fluid flow [48,87,91–95].

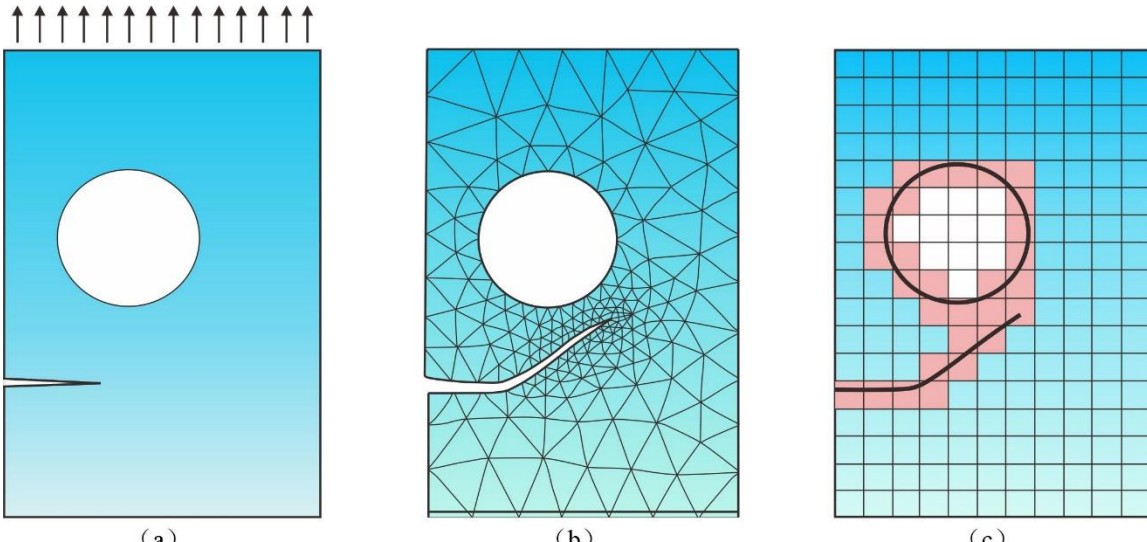

**Figure 4.** Differences between the traditional finite element method and the extended finite element method when modeling [95]: (**a**) Crack propagation in a flat plate containing holes, (**b**) Finite element meshing using mesh reconstruction method, (**c**) XFEM with enhanced technology.

A series of hydraulic fracture extend finite element models have been developed to achieve dynamic characterization of the hydraulic fracture propagation process in terms of the interaction mechanism between hydraulic fracture and natural fracture, fluid–solid

coupling, and fluid flow within the fracture. Wang et al. [95] established a two-dimensional hydraulic fracture extend finite element model by introducing a fracture tip enhancement function to realize the characterization of the dynamic process of fracture propagation in orthotropic anisotropic rocks. Wang [96] established a two-dimensional extend finite element model with fluid–solid coupling to simulate the fracture propagation behavior in orthotropic anisotropic rocks by considering rock deformation and fluid flow in the fracture and reached similar conclusions. All of the above studies are for the propagation of hydraulic fractures in two-dimensional planes, and some scholars have also studied the propagation of hydraulic fractures in three dimensions in space. By introducing a new solution algorithm in ABAQUS software, Liu et al. [44] achieved the arbitrary propagation of cracks in three dimensions in the present extended finite element model. Dehghan et al. [97] studied the interaction between natural fractures and hydraulic fractures in three dimensions based on the extended finite element module in ABAQUS software, and that the horizontal stress difference and the orientation and dip angle have important effects on the propagation and geometry of hydraulic fractures. Although the 3D model can characterize the fracture propagation in more detail, due to the limitation of technology, the currently established 3D extended finite element model cannot characterize the hydraulic fracture propagation process efficiently, and the cost of the 3D model is also high, so it is mainly based on the extended finite element to establish a 2D model to simulate the hydraulic fracture dynamic propagation process. Although the extended finite element method is superior in dealing with fractures.

### 2.2.3. Boundary Element Method

The boundary element method belongs to the same continuum media mechanics method as the finite element method [13,14,17,32,71,92,98]. However, unlike the finite element method, which directly divides the cells of the whole system, the boundary element method only divides the cells at the boundary or discontinuous interface of the defined domain, takes the boundary integral equations defined at the boundary as the controlling equations, and approximates the boundary conditions with functions satisfying the controlling equations to solve the system of equations numerically [13,14,17,71,98]. Since the boundary element method only divides the cells on the boundary, the number of cells and the amount of data required to divide the discrete problem using this method is relatively small, thus significantly improving the simulation efficiency. When solving the hydraulic fracture propagation process using the boundary element method, the fracture needs to be discretized by adding a few cells to the newly created fracture tip and then using the control equations to approximate the boundary conditions, thus characterizing the dynamic propagation process of the fracture [13,14,17,71,98].

Many scholars have introduced the boundary element method into the study of the dynamic propagation mechanism of hydraulic fractures and have achieved some understanding of the fracture propagation mechanism. Among them, the displacement discontinuity model established by Crouch et al. [98] based on the boundary element method for the first time can reduce the dimensionality of the computational model, reduce the number of grids, and improve the computational efficiency, so it is widely used in the study of dynamic extension of hydraulic fractures. Xie et al. [99] established DDM to study the behavior of hydraulic fracture turning to natural fracture and the geometric shape of hydraulic fracture, and found that the deflection of hydraulic fracture to natural fracture is controlled by many factors, and the shape of fracture is also complex. Cheng et al. [100] also investigated the effect of natural fractures near the borehole on hydraulic fracture propagation using DDM and pointed out that the pre-existing fracture length has a significant effect on the rupture pressure. Ren et al. [101] corrected DDM by using the results of physical model experiments and introducing propagation modes and criteria, and simulated and studied the effects of natural fracture distribution and construction parameters on crack extension using the modified model. It is found that the reasonable construction scheme and fracturing parameters are beneficial to the generation of complex

fracture networks. In addition to the analysis of the interaction between hydraulic fractures and natural fractures, Zhao et al. [102] initially established a three-dimensional DDM by introducing a fracture initiation and propagation criterion, and the model achieved the characterization of the geometry of complex fracture networks in shale gas reservoirs (Figure 5). Although the DDM has many advantages, it is currently used only as an auxiliary tool to calibrate the results of dynamic fracture propagation studies because the model does not consider interseam fluid flow and cannot simulate problems, such as hydraulic fracture deflection under the action of induced stress fields [52,69,103,104].

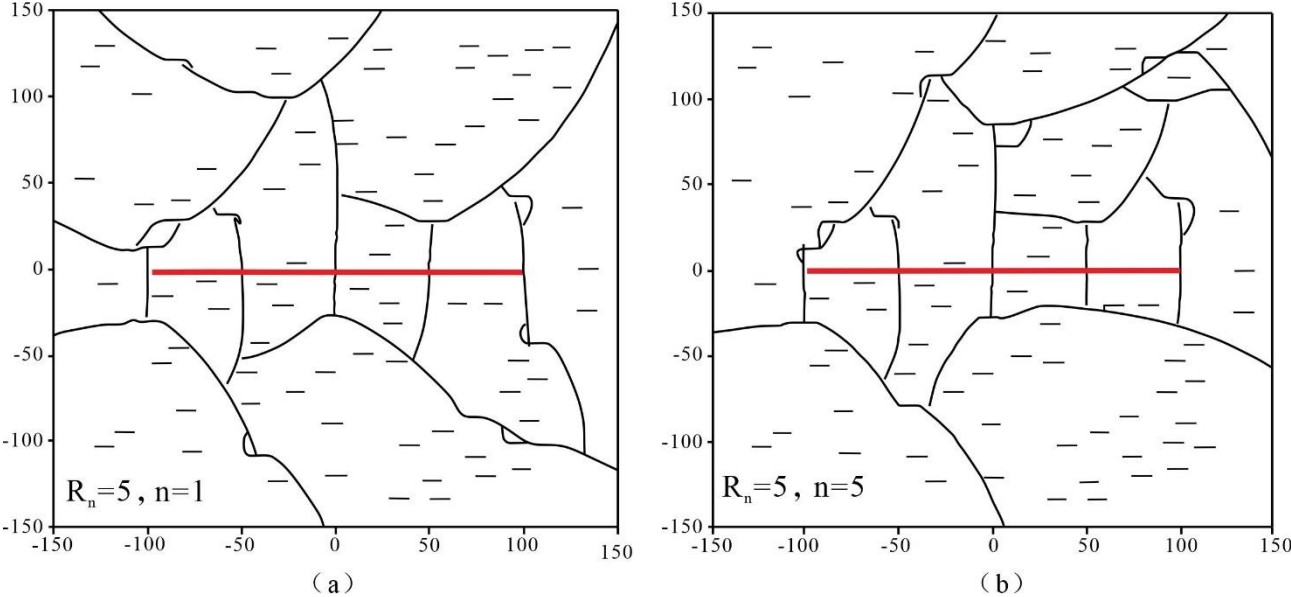

**Figure 5.** Two-dimensional discrete element model simulates the intersection of multi-stage fractured hydraulic fractures with natural fractures in horizontal wells, hydraulic fracture pattern for a sequential injection from left to right in the presence of 100 randomly located natural fractures with a 10 m initial length. There were 5 injection points, located at x = −100, −50, 0, 50 and 100 m and y = 0 m. The figure shows a bedding plane (map) view of the fracture geometry, and the fractures are confined to an elastic layer with a 30 m thickness. Both simulations were run with a relative net pressure of Rn = 5, but with different propagation velocity exponents—(**a**) n = 1, (**b**) n = 5 [52].

Currently, many scholars have conducted a lot of research on the fracture propagation problem in the hydraulic fracturing process and achieved certain results. The analysis found that to be able to finely characterize the fracture propagation process, the displacement discontinuity model based on the boundary element method has gradually developed from two-dimensional and proposed three-dimensional to three-dimensional [14,101], and the research direction has also evolved from single-section fracturing of horizontal wells to multi-section fracturing of horizontal wells, and the influence of factors, such as ground stress, natural fractures, fracture interaction and fracturing fluid on hydraulic fracture propagation, has been analyzed, and the main control factors affecting hydraulic fracture. Based on these insights and conclusions, the research method was improved to better characterize the propagation pattern of hydraulic fractures, and the cycle was repeated to finally understand the fracture propagation mechanism to optimize the fracturing process and to improve the reservoir transformation volume.

### 2.2.4. Discrete Element Method

The discrete element method was originally proposed by Cundall as a numerical calculation method based on the mechanics of discontinuous media [102–104]. The basic idea of this method [2,13,14,105,106] is to discretize the object of study into rigid blocks: each rigid block follows Newton's second law; each rigid block is connected by contact; the

interaction between the rigid blocks is described by using the display difference method; the discontinuous units formed between different rigid blocks can be used to describe weak surface structures, such as cracks and laminar surfaces; and the discontinuous units between the channels formed between the discontinuous cells allow for fluid flow, thus solving the fluid–solid coupling problem (Figure 6a). The discrete element method has been introduced by researchers to study the mechanism of hydraulic fracture extension because of its greater advantage in dealing with discontinuous structures, such as natural fractures and laminated surfaces [13,14,17,106,107].

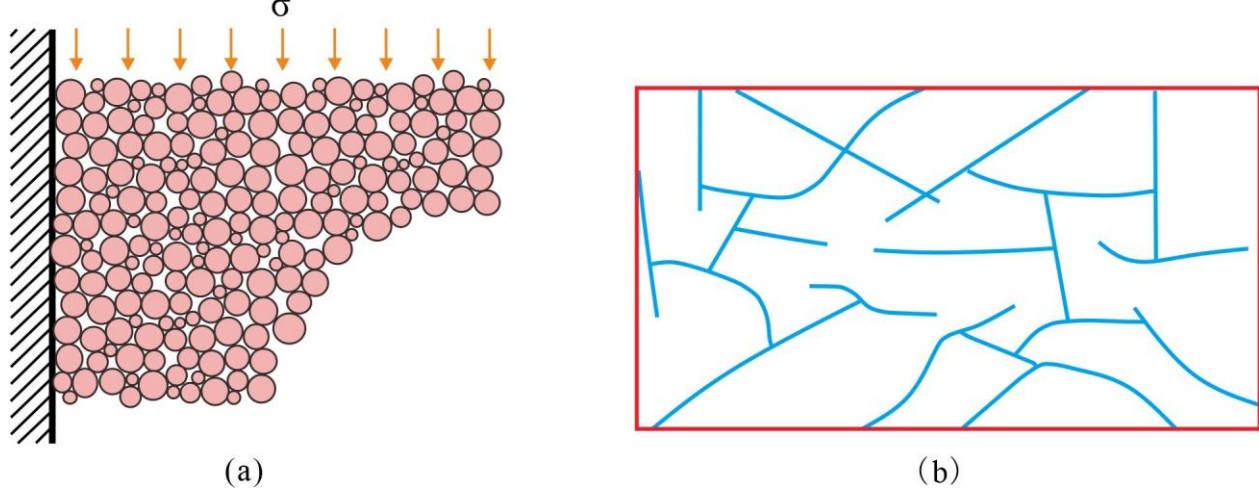

**Figure 6.** Simulation of hydraulic fracture propagation by discrete element method [28,108]: (**a**) Discrete particle distribution, (**b**) Unconventional fracture network distribution, blue represents cracks.

Many scholars have established discrete element models based on the discrete element method to simulate the dynamic propagation process of hydraulic fractures during the fracturing process, and among these models, the common models are the wireline network model, discrete fracture model, randomly distributed multi-fracture model and unconventional fracture model [2,14,17,32,109]. Among them, the discrete fracture model based on the discrete element method is the more widely used model, which can more accurately characterize the propagation pattern and geometry of hydraulic fractures by taking into account the effects of fluid filtration and fracture interaction during the fracturing process. Shen et al. [109] established a discrete element model to simulate the effect of conglomerate particles on the extension of hydraulic fractures. It was found that conglomerate particles affect fracture propagation in two ways, penetration and avoidance, and the penetration pressure of conglomerate strata is linearly and nonlinearly related to particle strength and content, respectively. Based on the advantages of the discrete element method in studying discontinuous structures, Lu et al. [110] established a discrete element model to simulate the effect of coal seam cuttings on hydraulic fracture propagation and found that hydraulic fractures mainly extend along the direction of maximum principal stress. In addition to the influence of the developmental characteristics of different rocks on hydraulic fracture propagation, other scholars have analyzed the influence of ground stress difference, fracture interaction, tectonic parameters, and other fundamental features on the hydraulic fracture propagation [111]. Tan et al. [112] established a discrete element model to analyze the effects of ground stress difference, tectonic parameters, and rock interface properties on hydraulic fracture propagation and found that the ground stress difference is the key factor controlling the fracture propagation path. In conclusion, scholars have conducted a lot of research on the influencing factors of hydraulic fracture propagation during rock fracturing using the discrete element method, and have gained some understanding of them. From the initial two-dimensional discrete element model, which gradually developed to the proposed three-dimensional and full three-dimensional models [2,14,17], the research work has achieved more results, and with continuous research, the understanding of the propa-

gation mechanism of hydraulic fractures has become more in-depth, so that the established discrete element model can more graphically characterize the dynamic propagation process of hydraulic fractures. However, for more complex crack networks, due to more influencing factors and more artificially set parameters in the simulation process, the simulation results have a large error, which is difficult to reflect in the real fracture network.

In addition, the unconventional fracture model has been favored by many scholars as an emerging model that can determine whether hydraulic fractures are captured by or directly through natural fractures and can simulate the interfering effects between hydraulic fractures (Figure 6b). The unconventional fracture model was first proposed by Weng et al. [16] based on the discrete element method to simulate the effect of natural fractures on the generation of complex fracture networks to study the effect of stress shadowing effect on fracture propagation, and it achieved a three-dimensional characterization of the three-dimensional complex fracture network during the fracturing process. In addition to modeling based on theoretical analysis, Haege et al. [19] also introduced micro-seismic monitoring technology into the modeling of unconventional fractures and used micro-seismic monitoring data to make corrections, thus establishing an unconventional fracture model to characterize the dynamic propagation process of hydraulic fractures during the fracturing process. However, since the model assumes that natural fractures and hydraulic fractures are perpendicular, it cannot simulate those curved hydraulic fractures and natural fractures, and most of the shales have more or less curved fractures in both natural and hydraulic fractures, so the results obtained by using this method to characterize the propagation of hydraulic fractures need further investigation. The results obtained using this method to characterize the propagation of hydraulic fractures need to be further validated [14,17,108,113]. However, for the bending fractures in shale reservoirs, the results obtained by this method will have a large deviation from the actual value. It is also necessary to improve this model according to the relevant characteristics of hydraulic fractures so that it can better represent the dynamic propagation process of fractures.

### 2.2.5. Phase Field Method

The phase field method [14,70,114,115] was originally proposed due to the difficulty of tracing the solid–liquid interface in solidification organization simulation and then it was improved and applied to the simulation of fracture initiation and propagation of hydraulic fractures during the fracturing process. The modified phase field method considers that the reservoir rock will be deformed elastically during the fracturing process due to the change of force and accumulated elastic energy, and when the accumulated elastic energy exceeds the critical value of the reservoir rock, the rock will be a failure and a fracture will be created, but the rock around the fracture will not be destroyed because the elastic energy does not exceed the critical value, thus forming a damage transition zone around the fracture. The phase-field method [14,17,116,117] is based on the generalized and variational theories for the simulation of fracture propagation, and the smooth transition between the shale matrix and the fracture is described by introducing a phase-field variable, and the fracture can propagate spontaneously with the evolution of the phase-field variable.

The phase field method has received much attention because of its ability to handle a large number of complex fracture networks simultaneously, and many scholars have introduced the phase field method to study the propagation law of hydraulic fractures. Li et al. [117] established a two-dimensional phase field model for studying the interaction between hydraulic fractures and natural fractures during the fracturing process. It was found that the hydraulic fracture always propagates along the minimum energy path during the propagation process, and the propagation of the hydraulic fracture is related to the stress anisotropy, injection rate, hydraulic fracture approach angle, etc. Liang et al. [118] established a new two-dimensional phase field model by improving the traditional phase field model and used it to simulate the effect of natural fractures on hydraulic fracture propagation in porous media. The simulation results show that small approach angles favor the deflection of the hydraulic fracture toward the natural fracture and activate the

natural fracture. In addition to the study of the hydraulic fracture propagation law, some scholars introduced new algorithms to improve the phase field model and to increase the solution rate of the model. Borden et al. [119] established a phase field model based on the high-order fracture density function, which greatly improved the convergence speed during the calculation. Almani et al. [120] introduced a multi-rate fixed stress splitting iteration in the phase field model coupling scheme to realize the coupling of flow and geomechanics in poroelastic media, thus substantially improving the computational efficiency. It can be found that since the phase field method has unparalleled advantages in the study of complex seam networks, more and more scholars have used this method to study the propagation law of hydraulic fractures and the generation of complex seam networks during the fracturing process and have achieved certain results. It is foreseeable that with the rise of unconventional oil and gas resources, such as shale oil and gas, the phase field method will become an important tool for scholars to study the mechanism of complex fracture network generation.

## 3. The Main Control Factor of Fracture Propagation in Shale Reservoir Fracturing

Scholars have conducted numerous studies on the main control factors affecting fracture propagation in fractured shale reservoirs using physical simulation experiments and numerical simulations. In general, the main controlling factors of fracture propagation in shale reservoirs [2,14,17,28,30,56,121,122] can be divided into two categories: geological factors and engineering factors. Among them, geological factors are caused by the nature of the shale reservoir itself, mainly the mineral composition of the shale, rock mechanical parameters, natural fractures, laminar surfaces, fracture interaction, and crustal stress [22,31,38,82,120,123,124]. The engineering factors, on the other hand, are mainly the effects of fracture propagation in shale reservoirs due to the artificially selected fracturing process in shale gas extraction engineering, the fracturing fluid, construction displacement, injection pressure, and injection rate [11,12,20,61,64,125,126]. Among them, geological factors are unchangeable, but the "sweet spot" areas with good geological conditions can be selected for shale characteristics, and priority can be given to their exploitation, while engineering factors are controlled by human factors, so the fracturing process can be optimized through the in-depth understanding of the previous geological studies to achieve efficient shale gas.

### 3.1. Geological Factor

During the fracturing and modification of shale, the mineral composition of shale is often classified into three main categories: brittle minerals, clay minerals, and others [127,128]. Among them, the content of brittle minerals is often used to evaluate the practicability of shales [129,130]. The higher the content of brittle minerals in shale, the more brittle and compressible it is, and the more favorable it is for fracturing [131–133]. In addition to characterizing the compressibility of shales using mineral composition, their compressibility can also be characterized using rock mechanical parameters [134–136]. Generally, the elastic modulus and Poisson's ratio of shale are commonly used to characterize its compressibility, and the larger the elastic modulus and the smaller the Poisson's ratio, the more compressible the shale. Meanwhile, fracture toughness [137–139] has an important influence on the propagation of shale fractures. It is shown that the stronger the fracture toughness of the shale, the weaker its brittleness and the lower its compressibility, i.e., the fracture toughness of the shale is inversely proportional to its brittleness. In addition to the composition and mechanical properties of the shale itself, the presence of discontinuous structures, such as natural fractures and laminae surfaces in the rock, also has a serious impact on the propagation of hydraulic fractures. The available research data show that the discontinuities such as natural fractures and laminated surfaces present in shale have a more complex effect on hydraulic fracture propagation, and when hydraulic fractures encounter discontinuities, such as natural fractures and laminated surfaces, they may pass directly through discontinuities, such as natural fractures and laminated surfaces, deflect

along natural fractures, bifurcate along natural fractures, and be cut off by natural fractures [2,12,60,140] (Figure 7), which is related to factors, such as horizontal stress difference and angle of approach. The smaller the horizontal stress difference and the smaller the approach angle of the shale, the easier it is for hydraulic fractures to propagate along discontinuities, such as natural fractures and laminae surfaces, causing discontinuities, such as natural fractures and laminae surfaces, to be reactivated [50,61,141,142]. The degree of influence of horizontal stress difference and approximation angle on the interaction between hydraulic cracks and natural cracks was studied quantitatively by combining indoor physical simulation experiments with numerical simulations. The simulation results found that when the horizontal stress difference was less than 8.0 MPa and the approach angle was less than 45°, the natural fracture was easily reactivated; when the horizontal stress difference was greater than 9.0 MPa and the approach angle was greater than 60°, the hydraulic fracture passed directly through the natural fracture, and the natural fracture was activated to the least extent at this time [17,140,143]. In contrast, the results of the indoor physical simulation experiments show that the hydraulic fracture passes through the natural fracture and continues to propagate along the direction of the maximum horizontal principal stress when the horizontal stress difference is 6.0 MPa and the approach angle is 45° [2,17,126,140,143]. By comparing the results of both studies, it was found that the simulation results did not completely match the experimental results and there were some errors. In addition, the cementation strength of the laminae [82,112,144–146] has an important influence on the hydraulic fracture propagation. Shales with high cementation strength will hinder the longitudinal propagation of fractures, while shales with weak cementation strength tend to propagate along the laminae, i.e., too strong or too weak cementation strength of laminae is not conducive to the formation of complex fracture networks, and the dip angle of laminae also has a certain influence on the propagation of hydraulic fractures.

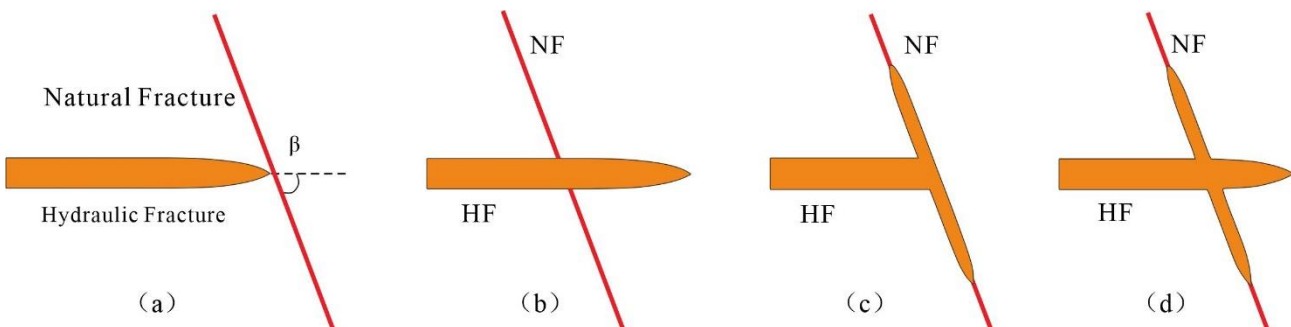

**Figure 7.** Results of the interaction between hydraulic cracks and natural cracks [117]: (**a**) Cut-off, (**b**) Penetration, (**c**) Deflection, (**d**) Bifurcate.

In addition to studying the propagation of a single fracture, the propagation of multiple fractures during fracturing [52] was also simulated. It is found that when multiple fractures are propagating, the fractures will interfere with each other, resulting in changes in the stress field, thus forming the stress shadow effect, which will interfere with the propagation of hydraulic fractures, resulting in complex fracture propagation morphology [2,17,52,85,147,148] (Figure 8). The stress shadow effect is particularly sensitive to the change of fracture spacing, with the feature that the stress shadow effect gradually increases with the decrease of fracture spacing [149,150]. In addition, the stress shadow effect is also related to horizontal stress difference, fracture pressure, and fracture propagation angle, and linearly increases with the increase of vertical stress. In the process of multi-fracture propagation, the stress shadow effect will cause hydraulic fractures to deflect easily, and a higher horizontal principal stress difference will inhibit the deflection of hydraulic fractures, which is conducive to the formation of main fractures since energy is mainly used for the formation of main fractures, natural fractures, bedding planes, and

other discontinuities are not activated, which reduces the volume of reservoir reconstruction and is not conducive to the formation of complex fracture net-works [150–152]. Interseam interference is one of the important factors affecting the propagation of hydraulic fractures. At present, the research on the influence of stress shadow effect on fracture propagation in the process of multi-fracture propagation is mainly based on numerical simulation. By presetting the fracture propagation path and making a series of conditional assumptions on the fracturing process, the influence of the stress shadow effect on hydraulic fracture propagation is simulated and studied on a small scale. However, because the simulation scale is small and there are many assumptions, it can not describe the small-scale fracturing process in detail, so it can not characterize the fracture propagation of the reservoir scale fracturing process. Therefore, it is also necessary to improve the numerical simulation technology and combine it with other methods to study the propagation mechanism of hydraulic fractures under the influence of fracture interaction, to improve the reservoir stimulation volume.

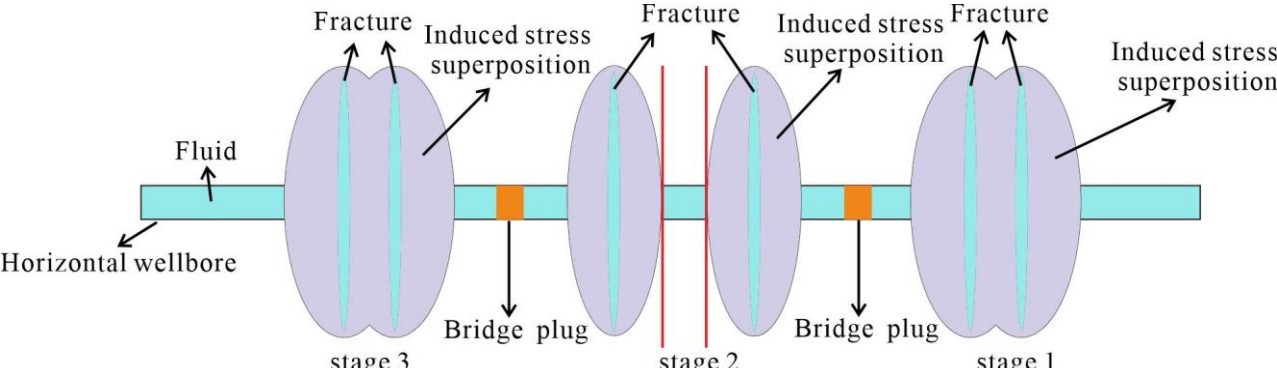

**Figure 8.** Influence of stress shadow effect caused by multi-stage fracturing in horizontal well on fracture propagation.

*3.2. Engineering Factor*

Unlike geological factors, engineering factors are determined by the operator, and one can optimize the fracturing process by setting reasonable construction parameters to form a complex fracture network, thus improving the reservoir stimulation effect. It has been found that the engineering factors affecting fracture propagation in shale reservoirs are mainly the type of fracturing fluid, construction pressure, injection rate, and other factors [153–158]. In hydraulic fracturing operations, different fracturing fluids have different viscosities and wettability [158,159], and the viscosity of fracturing fluid is related to the energy consumption during hydraulic fracture propagation; the lower viscosity of fracturing fluid loses less energy during fracturing, the more energy is used for fracture propagation, which is beneficial to hydraulic fracture propagation [10,63,160]; the wettability of fracturing fluid can reflect its disintegration The stronger the wetting of the fracturing fluid, the stronger its ability to disintegrate the rock, which is conducive to the hydraulic fracture communicating more natural fractures and forming a complex network volume [161,162]. The effect of construction parameters on hydraulic fracture propagation was investigated using physical experimental simulations, and it was found that a constant injection pressure could prolong the fracture propagation period, and it was favorable to generate complex, curved hydraulic fractures at a relatively high injection pressure, while it was easy to generate homogeneous, simple fractures at a constant injection rate [163]. Furthermore, the decrease in pumping pressure, the increase in injection rate, and the increased pressure decay rate can be used as the evaluation index of hydraulic fracture initiation and propagation. In addition, the injection method, fracturing sequence, and proppant [64,94,164] also have different degrees of influence on the propagation of hydraulic fractures, and when analyzing the engineering factors affecting hydraulic fracturing, the appropriate fracturing process should be selected according to the geological characteristics of the shale and combined with the relevant

fracturing process for the relevant parameters of the shale, and the in-fluence of each factor in the development process should be comprehensively analyzed to clarify the degree of influence of each engineering factor to better guide the fracturing design and form a complex fracture network to improve the reservoir stimulation effect.

To sum up, the influencing factors of hydraulic fracture propagation are multifaceted and complex. When studying the influencing factors, we should first analyze the influence of single geological factors and multiple geological factors on hydraulic fracture propagation from the geological perspective, classify their influence degrees, and identify the main controlling factors. Then, from the aspect of engineering development, the influence of various parameters on hydraulic fracture propagation during fracturing construction is analyzed, and the order is made according to the degree of influence so as to identify the main control factors. Finally, the geological influence factors and engineering influence factors are combined, and the influence of each factor and main control factors on the propagation of hydraulic fractures is explored from the perspective of fracturing construction so as to characterize the dynamic propagation process of hydraulic fractures in detail so as to optimize the fracturing process, improve the reservoir reconstruction effect and obtain higher shale gas production.

## 4. Discussion

(1) Currently, physical simulation experiments and numerical simulation techniques are often used to study the propagation mechanism of hydraulic fractures. Among them, the indoor physical simulation experimental method uses small-sized rock samples for hydraulic fracture tests, which simulates the fracturing operation process at the subsurface reservoir scale indoors and can reflect the propagation of hydraulic fractures during the fracturing process in a simple and intuitive way [58,60,144,165]. In addition, indoor physical model experiments can be used to explore the influence of one or some parameters on the hydraulic fracture propagation pattern and to analyze the main control factors affecting the fracture propagation. Therefore, this method has become a common method for scholars to study the mechanism of hydraulic fracture propagation. However, due to the complex and variable subsurface conditions, especially for deep shale reservoirs with large burial depths, it is difficult to provide a realistic picture of the actual fracturing process by using indoor physical simulation experiments. In addition, the reliability of the test results is also questionable when using such a small-scale hydraulic fracturing test to characterize the fracturing operation at the reservoir scale due to the small scale of the samples used in the indoor experiments and the large scale of the subsurface reservoir [10,72,166,167]. Therefore, when indoor physical simulation experiments are used to study hydraulic fracturing at reservoir scale, the results also need to be evaluated, and currently, auxiliary techniques, such as acoustic emission monitoring, CT scanning, and post-compression morphology observation [11,18,21,168], are often used to correct the experimental results of indoor hydraulic fracturing. However, because the accuracy of acoustic emission monitoring technology still needs to be improved, and the high cost and limited image content of CT scanning technology also fail to verify the results of hydraulic fracture simulation exactly, effective technical methods need to be developed to finely characterize the dynamic propagation of hydraulic fractures during fracturing so as to identify the propagation mechanism of hydraulic fractures and better guide the fracturing construction;

(2) In addition to indoor physical simulation experiments, numerical simulation techniques are also commonly used to study the hydraulic fracture propagation law. Compared with indoor physical simulation techniques, numerical simulation techniques have the advantages of low cost, short time consumption, and quantification, so they become the first choice for studying the hydraulic fracture propagation law in the hydraulic fracturing process. Currently, the commonly used numerical simulation methods include the finite element method, extended finite element method, boundary element method, discrete element method, phase field method, etc. Each numerical simulation method has its advantages and shortcomings (Table 1), and the results obtained by different numerical simulation

methods are not consistent when using these numerical simulation methods to study the fracture propagation law of the same re-search object. Therefore, it is necessary to select appropriate numerical simulation methods and their combinations to investigate the fracture propagation mechanism for the relevant characteristics of shale reservoirs. In studying the expansion law of hydraulic fractures using numerical simulation techniques, a series of assumptions need to be made for the target reservoir so that the complex characteristics of reservoir proper-ties, fluid motion, and geomechanics [14,17,49,82] can be simplified so that the relevant mathematical models can be established and the established models can be solved using numerical methods to reveal the expansion mechanism of hydraulic fractures. However, due to the simplification of the subsurface temperature and pressure conditions, reservoir properties, and fluid motion within the formation, there is a problem with approximation between the results obtained using this method to characterize the propagation pattern of hydraulic fractures in real fracturing operations, i.e., the results obtained using the numerical simulation method can be seen as an approximation of the actual results, but the approximation remains to be further verified. Therefore, although the numerical simulation method has many advantages, it fails to achieve a fine characterization of the hydraulic fracture expansion process when the expansion mechanism of hydraulic fractures is studied using this technique because the simulation results, similar to the indoor physical model experiments, also need to be verified. Therefore, numerical simulation methods need to be studied and improved to better reveal the propagation mechanism of hydraulic fractures;

(3) When studying the propagation law of hydraulic fractures during fracturing, it is difficult to accurately characterize the dynamic propagation of hydraulic fractures during fracturing, regardless of whether indoor physical simulation experiments or numerical simulation methods are used. There is an urgent need for a method that can effectively characterize the hydraulic fracture propagation law and its influencing factors to reveal the dynamic propagation characteristics of hydraulic fractures. Based on this, scholars have introduced a series of auxiliary techniques, such as acoustic emission monitoring, CT scanning, and post-pressure morphological observation, into the study of hydraulic fracture propagation law and used the auxiliary techniques to correct the experimental and simulation results, which greatly improved the reliability of the results. However, these auxiliary techniques also have their shortcomings, such as the limited accuracy of acoustic emission monitoring technology, CT scanning's difficulty carrying out large-scale applications due to the small imaging area, and because post-pressure morphological observation can only rely on indoor physical model experiments, there is a need to strengthen the improvement of auxiliary techniques to improve their applicability to the study of fracture propagation law;

(4) Some scholars have combined all the methods for characterizing crack propagation, selected the corresponding combination of numerical simulation algorithms according to the characteristics of different moments and regions of crack propagation, and combined them with indoor physical model experiments and a series of auxiliary techniques to establish a research method for characterizing the crack propagation law. This method can substantially improve the computational accuracy and better characterize the crack propagation law. Therefore, in addition to improving various research methods to characterize fracture propagation more accurately, it is also a good research direction to combine them to characterize hydraulic fracture propagation during fracturing according to the advantages and disadvantages of each research method, but the results obtained from this method also need to be verified. In particular, the joint use of various numerical simulation methods has become a new trend in the study of hydraulic fracture propagation law. Scholars have combined the advantages of various numerical simulation methods, discrete unitization of the research object, and selected suitable numerical simulation methods for different units to characterize the dynamic propagation process of hydraulic fractures, which further reveals the propagation law of hydraulic fractures and deepens scholars' understanding of the propagation mechanism of hydraulic fractures. Therefore, for the current techniques

used to study the propagation law of hydraulic fractures, although each has its advantages, there are certain shortcomings in either method, and the combination of various re-search methods and some auxiliary techniques to characterize the propagation law of hydraulic fractures is favored by the majority of scholars and is also the mainstream direction for the future study of the propagation law of hydraulic fractures.

**Table 1.** Effect of stress shadow effect on multi-crack propagation [14,17,45,82,85,86,88,100,119].

| Research Method | | Advantages | Disadvantages |
|---|---|---|---|
| Indoor physics simulation experiments | | Enables miniaturization of hydraulic fracturing operations; allows for investigation of the effect of single or multiple parameters on hydraulic fracture propagation patterns. | Difficult to reflect the actual hydraulic fracturing operation process; the reliability of experimental results needs further verification; high cost. |
| Numerical simulation | Finite Element Method | The grid division is flexible and suitable for cracks with irregular boundary shapes, while the accuracy is high and convergence is good. | The high stress area near the crack tip requires a large number of meshes; the simulation of fracture propagation requires a large amount of machine time to re-dissect the mesh. |
| | Extended Finite Element Method | The crack surface does not need to overlap with the cell boundary, and no mesh reconstruction is required when the crack propagates along an arbitrary path, greatly reducing the computational effort. | Using horizontal set of virtual nodes to realize intersection and bifurcation of cracks, the extended simulation of complex cracks becomes very complicated. |
| | Boundary element method | Only the boundary of the discrete solution domain reduces the degrees of freedom and the amount of original information of the problem; it is suitable for solving infinite domain problems; it has higher accuracy and reliability. | It is difficult to apply for problems such as non-uniform media; for general nonlinear problems, the advantages of the boundary element method of discrete boundaries only are partially offset by the presence of the in-domain integral term in the equation. |
| | Discrete element method | The simulation of hydraulic fractures intersecting with natural fractures to form complex seam networks has a major advantage; the quasi-static and dynamic behavior of complex systems can be simulated without too many assumptions, using simple equations of motion. | The crack propagation direction is restricted; the computational volume is large, the memory occupation is large, the continuum problem cannot be well simulated, and the elastic homogeneous and grid-independent crack propagation cannot be calculated. |
| | Phase field method | Solving crack initiation propagation, branching and merging in a unified framework without additional standard or numerical processing is particularly convenient for simulating crack steering and crack intersection problems. | The computational cost is high, and for models with relatively large macroscopic dimensions (relative to the crack width), a very fine mesh needs to be used to solve them very accurately; the phase field parameters are not easily determined. |

## 5. Conclusions

This paper analyzes and summarizes the research methods and main control factors of hydraulic fracture expansion law during fracturing of shale reservoirs and summarizes the following conclusions on the current research status of hydraulic fracture expansion law during fracturing of shale reservoirs.

(1) At present, indoor physical simulation experiments and numerical simulation methods are mainly used to simulate the expansion law of hydraulic fractures during the fracturing process, combined with acoustic emission monitoring, CT scanning, and post-fracture morphology observation;

(2) Numerical simulation methods, such as finite element method, extended finite element method, boundary element method, discrete element method, and phase field method, are mainly used to study the hydraulic fracture expansion law;

(3) The relevant models established based on numerical simulation methods are the main means to carry out the study of hydraulic fracture expansion law. It mainly includes a finite element model based on the finite element method, an extended finite element model based on the extended finite element method, a displacement discontinuity model based on the boundary element method, an unconventional fracture model based on the discrete element method, etc.;

(4) The influencing factors of hydraulic fracture expansion in the fracturing process of shale reservoirs can be divided into two categories: geological factors and engineering factors. Among them, geological factors mainly include the mineral composition of shale, rock mechanical parameters, natural fractures, laminar surface, crack interaction, ground stress, and other factors. The engineering factors include fracturing fluid type, construction displacement, injection pressure, injection rate, etc.

**Author Contributions:** Conceptualization, X.G., Y.L., X.M.; investigation, X.G., Y.L.; writing—original draft preparation, X.G.; writing—review and editing, X.G., Y.L., G.L.; project administration, Y.L., X.M; funding acquisition, Y.L., X.M. All authors have read and agreed to the published version of the manuscript.

**Funding:** This research was funded by the project of R&D Department of Petrochina (No. 2021DJ2005).

**Data Availability Statement:** Relevant data are contained in the cited articles and reports.

**Acknowledgments:** The work was supported by the funding of the project of R&D Department of Petrochina (No. 2021DJ2005). Meanwhile, the author acknowledges the editor and reviewers for their constructive feedback to improve the quality of this publication.

**Conflicts of Interest:** The authors declare no conflict of interest.

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
