# Peer review of "Advances in Hydraulic Fracture Propagation Research in Shale Reservoirs"

_minerals, doi:10.3390/min12111438_

Round 1

Reviewer 1 Report

This paper discusses the advance of hydraulic fracture propagation research in shale reservoirs. The authors give sufficient and detailed discussions on hydraulic fracture propagation based on the principles and research results of indoor physical simulation methods and numerical simulation methods. It could be accepted after the following minor revision.

-The contents of the table should be further summarized and simplified. 

-The citation of Figure 7 should be added.

Reviewer 2 Report

Dear Authors,

I have gone through the manuscript and found it quite interesting and could be a good addition to the existing scientific literature related to shale reservoirs.

I have a few important suggestions to improve the quality of the manuscript.

1. Try to reduce the text in the Introduction and Methodology segment, otherwise, The submitted manuscript may appear as a report/thesis.

2. Why such studies are important for reservoir evaluations. To respond to this question, I would suggest citing the following studies.

a) Application of 3D static modeling for optimal reservoir characterization (2019). J Afr Earth Sci 152:184–196

b) Integrated wireline log and seismic attribute analysis for the reservoir evaluation: A case study of the Mount Messenger Formation in Kaimiro Field, Taranaki Basin, New Zealand. J. Nat. Gas Sci. Eng. 2022, 99, 104452

c) Reservoir description and characterization of Eni field offshore Niger Delta, southern Nigeria. J Pet Explor Prod Technol 8(2):381–397

d)Murree Clay Problems and Water-Based Drilling Mud Optimization: A Case Study from the Kohat Basin in Northwestern Pakistan. Energies 202215, 3424. https://doi.org/10.3390/en15093424

The table added to the manuscript carries too much of words. Please reduce the text in the table.

I would suggest minor revisions and would suggest incorporate the reviews into the manuscript.

Good luck

Reviewer 3 Report

Journal: Minerals

Manuscript ID: minerals-2006418

Title: Advance of Hydraulic Fracture Propagation Research in Shale Reservoirs

Authors: Xun Gong, Xinhua Ma, Yuyang Liu, Guanfang Li

Recommendation: Publish after minor revisions

Comments:

This paper discusses an overview the research methods of hydraulic fracture propagation in shale reservoir fracturing stimulation, based on the principles and research results of laboratory physical simulation methods and numerical simulation methods, provides a comprehensive analysis of the applicability, advantages, and disadvantages of various methods, and analyzes the principles, applicability and research progress of the more commonly used mathematical models established based on numerical simulation methods;  the influencing factors of hydraulic fracture propagation are discussed from both geological and engineering aspects, and the main controlling factors affecting fracture propagation are clarified.

The paper is clearly written and the result is good. The problem considered in the article is very urgent.

The article is written at a high scientific level. All references cited are valid.

I have some recommendations to improve the paper.

1.      Design of hydraulic fracturing laboratory experiments that capture similar phenomena to those expected at the field scale requires consideration of the scaling laws intrinsic to the mathematical model. Performing laboratory scaled tests on physical models to investigate processes that can occur at scales several orders of magnitude greater is an extremely useful tool for research, if the essential features of the process are captured and appropriate considerations are made when attempting to scale-up the experimental observations. In the last decade or so, several researchers have worked extensively on deriving scaling laws that relate physical model test parameters for hydraulic fracture propagation to the same processes occurring in subsurface wellbores. Failure to consider scaling effects in the design and interpretation of experimental work on hydraulic fracturing may conflict with observations on the importance of certain parameters that describe the process for field-scale fractures (de Pater, et al., 1994a (SPE 24994); Bunger, et al., 2005 (ARMA/USRMS 05-818)). This issue needs to be discussed in section 2.

2.      The review should include links to studies by Russian scientists on the development of hydraulic fracturing simulators on the Planar3D model.

 For example:

Article is devoted to mathematical modeling of the fracturing mechanics and software development for fracturing simulation and decisionmaking support in design and conduct of fracturing treatments. We discuss the basic software elements for modeling hydraulic fracturing, the existing mathematical models of hydraulic fracturing process (KGD, PKN, Radial, Cell-based-Pseudo3D, Lumped-Pseudo3D, Planar3D), history of development, characteristics and limitations. It is noted the practical importance for the fracturing planning tasks and risk minimization to correctly describe the fracture height growth, this pushed the development of Pseudo3D (P3D) and Planar3D (PL3D) models. (A.V. Aksakov et al., OIL INDASTRY, 2016, November. Corporate fracturing simulator: from a mathematical model to the software development)

The work is devoted to the validation of a Planar3D hydraulic fracturing model.. Hydraulic fracturing simulator is specialized software for mathematical modeling and engineering analysis of the hydraulic fracturing process. The simulator allows evaluating fracture geometry and treatment parameters, taking into account geological structure of the reservoir, rock geomechanical properties, fracturing fluid and proppant properties. Fracture model validation was carried out by comparing the results of mathematical modeling with the results of experimental studies in the laboratory installation of organic glass. (A.V. Pestrikov et al., OIL INDASTRY, 2016, November. Validation of the Planar3D hydraulic fracture model implemented in the corporate simulator RN-GRID)

Other works can be found by Planar3D.

Anyway, I recommend it for publication.

Additional Questions:

Originality: Good

Technical Quality: Good

Clarity of Presentation: Good

Importance to Field: Excellent
